# The Burden of Streptococcus pneumoniae-Related Admissions and In-Hospital Mortality: A Retrospective Observational Study between the Years 2015 and 2022 from a Southern Italian Province

**DOI:** 10.3390/vaccines11081324

**Published:** 2023-08-04

**Authors:** Fabrizio Cedrone, Vincenzo Montagna, Livio Del Duca, Laura Camplone, Riccardo Mazzocca, Federica Carfagnini, Valterio Fortunato, Giuseppe Di Martino

**Affiliations:** 1Hospital Healthcare Management, Local Health Autority of Pescara, Via Renato Paolini, 65124 Pescara, Italy; livio.delduca@asl.pe.it (L.D.D.); federica.carfagnini@asl.pe.it (F.C.); valterio.fortunato@asl.pe.it (V.F.); 2Postgraduate School of Hygiene and Preventive Medicine, Università Politecnica delle Marche, 60100 Ancona, Italy; s1092987@pm.univpm.it; 3Postgraduate School of Hygiene and Preventive Medicine, University of L’Aquila, 67100 L’Aquila, Italy; laura.camplone@graduate.univaq.it (L.C.); riccardo.mazzocca@graduate.univaq.it (R.M.); 4Department of Medicine and Ageing Sciences, “G. d’Annunzio” University of Chieti-Pescara, 66100 Chieti, Italy; giuseppe.dimartino@unich.it; 5Unit of Hygiene, Epidemiology and Public Health, Local Health Authority of Pescara, 65100 Pescara, Italy

**Keywords:** *Streptococcus pneumoniae*, hospital discharge records, in-hospital mortality, infectious disease, epidemiology

## Abstract

*Streptococcus pneumoniae* (SP) has high worldwide incidence and related morbidity and mortality, particularly among children and geriatric patients. SP infection could manifest with pneumonia, bacteremia, sepsis, meningitis, and osteomyelitis. This was a retrospective study aimed at evaluating the incidence, comorbidities, and factors associated with in-hospital mortality of pneumococcal disease-related hospitalization in a province in southern Italy from the years 2015 to 2022. This study was performed in the Local Health Authority (LHA) of Pescara. Data were collected from hospital discharge records (HDRs): this database is composed of 288,110 discharge records from LHA Pescara’s hospitals from 2015 to 2022. *Streptococcus Pneumoniae*-related hospitalizations were about 5% of the hospitalizations; 67% of these were without comorbidities; 21% were with one comorbidity; and 13% were with two or more comorbidities. Regarding mortality of SP infection, the most affected age group was older people, with the percentage of cases among the over-65s being more than 50% compared to the other age groups. HDRs represent a valid and useful epidemiological tool for evaluating the direct impact of pneumococcal disease on the population and also indirectly for evaluating the effectiveness of vaccination strategies and directing them.

## 1. Introduction

*Streptococcus pneumoniae* is one of the main causes of invasive and non-invasive human infectious diseases, with high worldwide incidence and related morbidity and mortality, particularly among children and geriatric patients [1]. The most common manifestation of pneumococcal disease is pneumonia, which represents one of the most frequent causes of community-acquired pneumonia (CAP). However, a wide range of clinical manifestations can occur due to *Streptococcus pneumoniae* infection. While some of these infections can be less serious, such as otitis, sinusitis, and bronchitis, others can be very dangerous and lead to illnesses such as bacteremia, sepsis, meningitis, and osteomyelitis. In these cases, we refer to these manifestations as invasive pneumococcal disease (IPD) [2].

The incidence, prevalence, and mortality due to pneumococcal diseases vary widely, both in different countries and over time, likely in relation to improved knowledge and management of the related diseases and principally to the implementation of universal vaccination campaigns on a national basis [3]. Invasive pneumococcal disease (IPD), defined as all clinical phenotypes in which *Streptococcus pneumoniae* was isolated from blood, cerebrospinal fluid, pleural fluid, or any other normally sterile site by culture, antigen testing, or molecular assays, was significantly reduced after the implementation of pneumococcal conjugate vaccines (PCV) [4]. The major risk factors for invasive pneumococcal disease include age (with highest incidence among children aged less than 2 years and the elderly aged over 65 years), ethnicity, concomitant chronic illnesses such as diabetes, immunosuppressive drug therapy, and attendance of day care centers or community centers [5]. In Europe, the reported burden of disease varies across region as shown by the broad range of incidence of IPD and of antibiotic resistance of *Streptococcus pneumoniae*, particularly among children by country [6]. Although 91 distinct pneumococcal serotypes varying in capsular structure have been described and categorized into 46 different serogroups, not all are known to cause disease [7,8]. The incidence of IPD strongly decreased worldwide after the introduction of pneumococcal conjugate vaccines (PCV). In fact, mass vaccination campaign remains the most important public health strategy that strongly impacts the burden of pneumococcal diseases, by reducing the incidence of IPD and pneumococcal infections. Two types of vaccines are available worldwide: a pneumococcal conjugate vaccine (PCV), initially recommended for infant immunization, and a pure polysaccharide pneumococcal vaccine (PPV), recommended for adult immunization after a prior dose of PCV. Several countries have recommended pneumococcal vaccination in both children and older adults as the best strategy for reducing the circulation of *Streptococcus pneumoniae* in these populations [9]. In 2012, with the approval of the Italian National Vaccine Prevention Plan (PNPV 2012–2014), the pneumococcal vaccine was introduced into the national vaccination schedule. Pneumococcal vaccination with the 13-valent conjugate vaccine is recommended and offered free of charge to all newborns, with three doses in the first year of life, and to individuals of any age with conditions at risk that increase the likelihood of serious complications. Vaccination is also recommended and free for all individuals aged 65 and over, regardless of the presence of particular risk situations. For newborns, simultaneous administration with the hexavalent vaccine is recommended. In elderly people and individuals at risk, vaccination with a first dose of PCV13 is recommended, followed by vaccination with the polysaccharide vaccine PPSV23. Preventing the occurrence of pneumococcal disease is of paramount importance as it can lead to a healthier population, help combat antimicrobial susceptibility, and reduce the cost of hospitalization for pneumococcal disease.

In fact, the burden of *Streptococcus pneumoniae* has also a strong economic impact [10], increasing medical costs due to access to healthcare, visits, and hospitalizations.

In the United States, the total annual cost for the adult population over 50 years old is 3.7 billion dollars. In Italy, it was estimated that the yearly cost for treating a patient with CAP, including the costs of the follow-up period, was EUR 1586.7 [11].

However, despite these recommendations, vaccination coverages in Italy are low among adults.

Furthermore, data on the real burden of pneumococcal diseases are poor due to the frequent lack of cultural identification of the causal agent during the medical care of the disease. As a consequence, among upper respiratory tract infections, the etiologic fraction attributable to *Streptococcus pneumoniae* is strongly underestimated [9].

The mortality rate of IPDs in Europe is 15%. However, the mortality rate varies among different age groups, with 4% in children under 15 years of age, 6% in 15–44-year-olds, 11% in 45–64-year-olds, and 21% in those who are over 65 years old. Therefore, the mortality rate increases with age. According to the latest report on invasive bacterial diseases from the Italian Institute of Health (ISS—Istituto Superiore di Sanità), the incidence of invasive pneumococcal disease in Italy was 2.77 cases per 100,000 population in 2019 [3].

For these reasons, according to the World Health Organization (WHO), pneumococcal disease is a major public health problem worldwide. It is estimated that approximately one million children die from pneumococcal disease every year [2].

In Italy, hospital discharge records (HDRs) are a useful tool to evaluate the burden of several diseases related to cost and healthcare utilization [12,13].

It included information on patients’ demographic characteristics and the diagnosis-related group (DRG) used to classify the admission and patients’ comorbidities, coded by ICD-9 CM codes. hospital discharge records (HDRs), despite some limitations, can be also considered a proxy for healthcare utilization. In particular, evaluating factors associated with healthcare utilization for patients affected by pneumococcal disease can lead to the improvement of preventive strategies at the regional or country level.

Poor studies were performed in Italy on pneumococcal disease using HDRs. In addition, the major part of them referred only to the pre-pandemic period. For these reasons, we conducted a retrospective study aimed at evaluating the incidence of pneumococcal disease-related hospitalization in a province in southern Italy from the year 2015 to 2022. In addition, we evaluated comorbidities and factors associated with in-hospital mortality.

## 2. Materials and Methods

This was a retrospective observational study performed in the Local Health Authority (LHA) of Pescara, a province of the Abruzzo region accounting for about 320,000 inhabitants. It has three hospitals: a tertiary referral hospital and two spokes. Data were collected from the LHA registry of hospital discharge records (HDRs). The HDRs include a large variety of data regarding patients’ demographic characteristics and hospitalization such as gender, ages and other information such as admission and discharge date and the discharge type, which also includes death. The HDRs also include information about diagnoses that led to hospitalization or that are concurrent including complications (a maximum of six diagnoses, one principal diagnosis and up to five comorbidities) and a maximum of six procedures or interventions that the patient underwent during hospitalization. Diagnoses and procedures were coded according to the International Classification of Disease, 9th Revision, Clinical Modification (ICD-9-CM), the National Center for Health Statistics (NCHS) and the Centers for Medicare and Medicaid Services External, Atlanta, GA, USA.

### 2.1. Coding of Streptococcus pneumoniae—Hospital Admission

For the selection of admissions with or without directly specified etiology, the following ICD-9-CM codes were used for the relative diagnoses:Pneumococcal Specified Pneumonias:
○481 (Pneumococcal pneumonia);○482.9 (Bacterial pneumonia, unspecified) and 041.2 (Pneumococcus infection in conditions classified elsewhere and of unspecified site);○485 (Bronchopneumonia, organism unspecified) and 041.2 (Pneumococcus infection in conditions classified elsewhere and of unspecified site);○486 (Pneumonia, organism unspecified) and 041.2 (Pneumococcus infection in conditions classified elsewhere and of unspecified site).Unspecified Pneumonias:
○482.9 (Bacterial pneumonia, unspecified);○485 (Bronchopneumonia, organism unspecified);○486 (Pneumonia, organism unspecified).Pneumonia All-cause:
○480.x, which includes the following diagnoses:
▪480.1 (Pneumonia due to adenovirus);▪480.2 (Pneumonia due to respiratory syncytial virus);▪480.3 (Pneumonia due to parainfluenza virus);▪480.8 (Pneumonia due to SARS-associated coronavirus);▪480.9 (Pneumonia due to other virus not elsewhere classified).○481 (Pneumococcal pneumonia);○482.x which includes the following diagnoses:
▪482.0 (Pneumonia due to *Klebsiella pneumoniae*);▪482.1 (Pneumonia due to Pseudomonas);▪482.2 (Pneumonia due to Hemophilus influenzae (*H. influenzae*));▪482.30 (Pneumonia due to Streptococcus, unspecified);▪482.31 (Pneumonia due to Streptococcus, group A);▪482.32 (Pneumonia due to Streptococcus, group B);▪482.39 (Pneumonia due to other Streptococcus);▪482.40 (Pneumonia due to Staphylococcus, unspecified);▪482.41 (Methicillin susceptible pneumonia due to *Staphylococcus aureus*);▪482.42 (Methicillin resistant pneumonia due to *Staphylococcus aureus*);▪482.49 (Other Staphylococcus pneumonia);▪482.81 (Pneumonia due to anaerobes);▪482.82 (Pneumonia due to escherichia coli (E. coli));▪482.83 (Pneumonia due to other Gram-negative bacteria);▪482.84 (Pneumonia due to Legionnaires’ disease);▪482.89 (Pneumonia due to other specified bacteria);▪482.9 (Bacterial pneumonia, unspecified).○483.x which includes the following diagnoses:
▪483.0 (Pneumonia due to *Mycoplasma pneumoniae*);▪483.1 (Pneumonia due to chlamydia);▪483.8 (Pneumonia due to other specified organism).○484.x which includes the following diagnoses:
▪484.1 (Pneumonia in cytomegalic inclusion disease,);▪484.3 (Pneumonia in whooping cough);▪484.5 (Pneumonia in anthrax);▪484.6 (Pneumonia in aspergillosis);▪484.7 (Pneumonia in other systemic mycoses);▪484.8 (Pneumonia in other infectious diseases classified elsewhere).○485 (Bronchopneumonia, organism unspecified);○486 (Pneumonia, organism unspecified);○487.0 (Influenza with pneumonia).Pneumococcal Specified Meningitis:
○320.1 (Pneumococcal meningitis);○320.2 (Streptococcal meningitis) and 041.2 (Pneumococcus infection in conditions classified elsewhere and of unspecified site);○320.82 (Meningitis due to Gram-negative bacteria, not elsewhere classified) and 041.2 (Pneumococcus infection in conditions classified elsewhere and of unspecified site);○320.9 (Meningitis due to unspecified bacterium) and 041.2 (Pneumococcus infection in conditions classified elsewhere and of unspecified site);○322.9 (Meningitis, unspecified) and 041.2 (Pneumococcus infection in conditions classified elsewhere and of unspecified site).Unspecified Meningitis:
○320.2 (Streptococcal meningitis);○320.82 (Meningitis due to Gram-negative bacteria, not elsewhere classified);○320.9 (Meningitis due to unspecified bacterium).Pneumococcal Specified Bacteriemia:
○038.2 (Pneumococcal septicemia (*Streptococcus pneumoniae* septicemia));○038.0 (Streptococcal septicemia) and 041.2 (Pneumococcus infection in conditions classified elsewhere and of unspecified site);○038.9 (Unspecified septicemia) and 041.2 (Pneumococcus infection in conditions classified elsewhere and of unspecified site);○790.7 (Bacteremia) and 041.2 (Pneumococcus infection in conditions classified elsewhere and of unspecified site).Unspecified Bacteriemia:
○038.0 (Streptococcal septicemia);○038.9 (Unspecified septicemia);○790.7 (Bacteremia).

In the case of unspecified pneumonia, meningitis, and septicemia, a specific percentage could be attributable to pneumococcal infection. According to the recent literature [13], for unspecified pneumonias the attributable percentage to SP could be 36%; for unspecified meningitis an attributable percentage to SP could be 58%; and for unspecified septicemias a percentage due to SP could be 20%.

The proportion of SP-HA was calculated on the assumption that all HDRs mentioning this pathogen were SP-HA, of the cases of pneumonia, meningitis, and septicemia for which no pathogen was specified.

### 2.2. Comorbidity Coding

Comorbidities were calculated according to Charlson through an algorithm proposed by Baldo et al. [14] which follows the ICD-9-CM codes. The comorbidities taken into account are previous myocardial infarction, peripheral vascular disease, cerebrovascular disease, dementia, chronic pulmonary disease, rheumatic disease, peptic ulcer disease, mild liver disease, diabetes without chronic complication, diabetes with chronic complication, hemiplegia or paraplegia, renal disease, any malignancy, moderate or severe liver disease, metastatic solid tumor, and AIDS/HIV.

### 2.3. Statistical Analysis

Qualitative variables were summarized as frequency and percentage. Annual admission rates for each SP-HA were calculated per 100,000 inhabitants using, when appropriate, the related attributable fractions, according to the most recent literature and as described previously [14].

The data related to the demographic structure, sex, and age of the population were collected through free access to the database on the website of the National Institute of Statistics (ISTAT).

Hospitalization rates were standardized for age and gender according to the Abruzzo population in the first year of the study (2015). To evaluate the association between in-hospital mortality and predictors, a multivariable logistic model was implemented using the presence or absence of death as the dependent variable (type of hospital discharge: death) and as independent variables, age expressed in categories (0–4, 5–14, 15–65, 65–79, and 80+), gender (M or F), the various invasive bacterial pathologies investigated (All Pneumonia, SP Pneumonia, Unspecified Pneumonia, SP Meningitidis, SP Bacteriemia, and unspecified Bacteriemia), and the presence of individual comorbidities according to Charlson (previous myocardial infarction, peripheral vascular disease, cerebrovascular disease, dementia, chronic pulmonary disease, rheumatic disease, peptic ulcer disease, mild liver disease, diabetes without chronic complication, diabetes with chronic complication, hemiplegia or paraplegia, renal disease, any malignancy, moderate or severe liver disease, metastatic solid tumor, and AIDS/HIV).

For all tests, a *p*-value less than 0.05 was considered significant. The statistical analysis was performed with STATA v14.2 software (StataCorp LLC, College Station, TX, USA).

## 3. Results

Our database comprises 288,110 discharge records from ASL Pescara’s hospitals covering the period from 2015 to 2022. *Streptococcus pneumoniae*-related hospitalizations numbered 14,506 (5.035%), of which 7906 (2.744%) were associated with invasive diseases, including 33 cases of meningitis (0.011%) and 88 cases of bacteremia (0.031%). In contrast, unspecified invasive infections accounted for 1673 pneumonia cases (0.581%) and 5 cases of bacteremia (0.002%). There were no diagnosed cases of meningitis without a defined etiology.

Patients were categorized into different age groups, and it was found that the majority of admissions occurred between the ages of 15 and 64 (42%). Hospitalizations were well distributed across all seven years, with a minimum of 30,166 in 2020 (10%) due to pandemic restrictions and a maximum of 39,225 in 2015 (14%).

Regarding patients’ comorbidity, we found that 192,088 had no comorbidity (67%), 59,573 had one comorbidity (21%), and 36,449 had two or more comorbidities (13%) according to the Charlson Index classification. In-hospital deaths totaled 13,434 (5%), with 3060 (1%) associated with *Streptococcus pneumoniae.* The sample is further detailed in Table 1.

About comorbidity distribution, apparently there was a similar pattern among cardiovascular, cerebrovascular, renal, and respiratory diseases and diabetes: the percentage of infants (0–4) with these comorbidities was higher than in children (5–14), and this fraction progressively increase with age (Figure 1): for instance, there were 4520 patients between 0 and 4 with at least one cardiovascular disease (9.63%), compared to only 41 patients between 5 and 14 (0.29%). The number of patients with cardiovascular comorbidities peaked in the oldest age class (over 80) with 8457 cases (22.79%). Similarly, cerebrovascular diseases were most common among those over 80 (5928 cases, 15.98%), decreasing to a minimum in children between 5 and 14, with 109 patients (0.78%). Cardiovascular, cerebrovascular, renal, and respiratory diseases and diabetes were more common among the younger age group than in the 15 to 64 age range.

Malignancies showed a slightly different pattern: cancer was most common between 65 and 79, with 9513 patients (13.89%). Malignancies were more frequently found in the central age class (15–64), followed by the younger one, with rates of 5.61% and 2.06%, respectively.

In-hospital mortality displayed a similar trend across all comorbidity classes (see Figure 1): it decreased from 0–4 to 5–14 and progressively increased for patients over 80.

Logistical analysis for in-hospital mortality (Table 2) confirmed the previously described trend: being younger than 5 and older than 64 is a risk factor for in-hospital mortality, with odds of 4.786 (*p* < 0.001) for ages 0 to 4, 2.868 (*p* < 0.001) for ages 65 to 79, and 6.599 (*p* < 0.001) for patients older than 80. Sex was not statistically significant as a risk factor (*p* = 0.480).

All S.P.-related invasive infections were correlated with in-hospital mortality, with the highest odds for *Streptococcus pneumoniae* (4.528 with *p* < 0.001), followed by S.P. meningitis (3.443 with *p* = 0.048) and lastly S.P. bacteremia (2.201 with *p* = 0.050). Among unspecified etiology infections, only pneumonias were significantly related to in-hospital mortality (7.098 with *p* < 0.001), while there was no association with bacteremia (*p* = 0.227). It was impossible to evaluate unspecified meningitis as a risk factor due to the lack of cases in the recorded seven years.

The great part of comorbidities included in our evaluation were significantly associated with in-hospital mortality, apart from COPD (*p* = 0.054), complicated diabetes (*p* = 0.267), and any -plegia (*p* = 0.160).

The estimated admission rate for all invasive S.P.-related infections in the Pescara province (Table 3a) reached its peak in 2019, with 501.1 cases per 100,000 people (95% CI 476.6–525.6). It gradually declined until 2022, with 361.4 admissions per 100,000 people (95% CI 340.7–382.0). The admission rate in 2022 was not significantly different from that in 2015, which was 343.3 per 100,000 people (95% CI 323.2–363.4).

The estimation for pneumonia associated with S.P. (Figure 2a) revealed a peak in 2019 (392.4 per 100,000 with CI 95% 370.7–414.0) but displayed a downward trend towards 2022, with 241.9 (CI 95% 225.0–258.7). Admission rates were significantly lower in 2022 compared to 2015, where it was 290.3 (CI 95% 271.8–308.8).

S.P. meningitis admission rate could not be estimated due to the lack of specified etiology.

Figure 2b showed no significant difference in overall trend, except between 2017 (3.1 per 100,000, CI 95% 1.2–5.1) and 2021 (0.3 per 100,000, CI 95% 0–1.0). In 2022, a not significantly higher admission rate was reported, with 2.2 admissions per 100,000 (CI 95% 0.6–3.8).

Figure 2c depicted the trend in S.P. bacteremia admission rate estimation, showing significant growth between 2018 (0.3 admissions per 100,000, CI 95% 0–0.8) and 2019 (7.6 per 100,000, CI 95% 4.5–10.6). However, the decrease in trend was not statistically significant, reaching 5.3 admissions per 100,000 (CI 95% 2.8–7.9) in 2022.The trend for in-hospital mortality related to all S.P.-related invasive infections (Table 3b) was lower in 2015 (61.3 per 100,000, CI 95% 52.8–69.9) compared to other years, with a significant increase in 2020 (150.9 in-hospital deaths per 100,000, CI 95% 137.6–164.2). Despite a significant lowering in 2022 (111.9 per 100,000, CI 95% 100.4–123.3), the current in-hospital mortality rate remains higher than in 2015–2018.

In-hospital pneumonia mortality rate showed significant growth in 2017 (73.1 per 100,000, CI 95% 63.7–82.4) compared to 2016 (54.5 per 100,000, CI 95% 46.4–62.6). For the other years, there were no significant differences, with the peak occurring in 2020, with 79.2 per 100,000 (CI 95% 69.5–88.9). In 2022, there were 57.9 in-hospital deaths associated with S.P. pneumonia per 100,000 inhabitants (CI 95% 49.7–66.2).

All S.P. meningitis-related in-hospital deaths occurred in 2018, with a rate of 0.6 per 100,000 (CI 95% 0–1.4).

S.P. bacteremia-related deaths began in 2018, with a rate of 0.3 per 100,000 (CI 95% 0–0.8), with no significant differences in the rate trend. In 2022, a not significantly higher in-hospital death rate has been reported with 0.6 per 100,000 (CI 95% 0–1.5), equal to the rate in 2021.

## 4. Discussion

With the present study, we analyzed HDRs, from 2015 to 2022, of a local health authority of Pescara, a province in southern Italy with three hospitals, two hubs and one spoke, and approximately 320,000 inhabitants. It was possible to evaluate the burden of hospitalization of all cases of pneumonia, pneumonia caused by streptococcus pneumonia, and pneumonia with non-specific causes. The study of HDRs has already been used as an indirect source of data to measure both the effectiveness of vaccination strategies [9].

Our data appear to be similar with the data of other works carried out in other Italian regions such as in Sicily [9] and the northeast of the country [14], with an admission rate percentage varying between 350 and 450 per 100,000 inhabitants. The epidemiological study of pneumococcal disease is of great importance because it can be effectively prevented through pneumococcal vaccination which has demonstrated its cost-effectiveness in different age groups of population [15,16].

The decrease in admissions observed during the years 2020–22 can be explained by the impact of pandemic on hospital admissions. Healthcare services focused their attention on COVID-19 patients during these years, causing on the other hand a decrease in admissions for other diseases, as reported in previous studies [17,18].

Regarding mortality, pneumonia causes over 27,000 deaths yearly across Europe [19]. We also calculated the odds of death due to invasive streptococcus pneumonia diseases. The most affected age group was older people, as expected, with a percentage of cases in the over-65s of more than 50% compared to the other age groups. The older age group is known to be the most affected group, and it shows the highest mortality risk associated with SP infection. It can be linked to a decrease in immune response and to the high frequency of co-morbidities among the elderly [9]. This point highlights that improving the vaccination among persons ≥65 may be the most cost-effective public health strategy in a community setting. Regarding factors associated with in-hospital mortality, cancers, dementia, heart failure, and kidney diseases are also known risk factors, in line with the previous literature [20]. The positive association of diabetes with mortality is controversial. Some studies reported a negative association [20]; other reported a simply non-significant association [21]. However, a hyperglycemic state caused by infections and relative treatments can worsen patients’ conditions, particularly among patients transferred to ICU [21,22]. However, diabetes is well known risk factor in 30- and 90-day mortality after discharge [21], but we are not able to obtain data on out-hospital mortality with this study. The similar age distribution of in-hospital mortality in patients with diabetes, CVDs, or renal diseases can be due to the most frequent distribution of these conditions in older age classes [12,17]. In addition, all these conditions are known risk factors for in-hospital mortality for many other medical conditions such as hip fracture, general surgery, and trauma [23,24,25].

Furthermore, from the analysis of the data from 2020, we reported a significant increase in the number of cases of pneumonia from all causes, compared with a constant or slightly decreased number of pneumonias from *Streptococcus pneumoniae*. This trend is probably compatible with the pandemic period, in which there was an increase in hospitalizations for COVID-19 pneumonia.

Also, the increased mortality rate reported in the year 2020 can be attributed to COVID-19. On the other hand, across all study periods, the mortality in the younger age class was negligible. This can be due to the extensive mass vaccination campaign performed, accordingly with the Italian National Vaccination Plan, that strongly reduced the mortality for PC diseases [9].

HDRs represent a very important source for invasive *Streptococcus pneumoniae* disease and for the evaluation of data concerning hospitalizations, comorbidities, and deaths. The study of HDRs has already been used as an indirect source of data to measure both the effectiveness of vaccination strategies [23,24,25] and to guide them [26,27,28].

The main strength of this study is the use of official, routinely collected electronic health databases from the entire population of an Italian province. To our knowledge, this is one of first study conducted in Europe covering a large study period (data from 7 years, from 2015 to 2022) and considering also the pandemic and post-pandemic periods. The evaluation of an entire stable population can be used as a proxy for the evaluation of primary prevention intervention, such as vaccination, or it can be considered as a useful tool to evaluate the impact of infectious diseases on hospital admissions. In addition, this is one of the first studies performed in the Abruzzo region on this topic. Another point of strength is the size of the HDR that we analyzed, which included 288,110 records. The large sample is useful to evaluate factors associated with hospitalization, making results generalizable.

However, this study has several limitations. Firstly, it represents the situation of a single province in southern Italy and the burden of the pneumococcal disease, which is a vaccine-preventable disease, and is therefore affected by the vaccination coverage that the local health authority has managed to achieve.

The second limit is that the HDRs were not completed with epidemiological intent but instead for remunerating purposes of admission. For this reason, comorbidities reported in each record can be overestimated or underestimated.

Thirdly, HDRs do not contain patients’ clinical data, such as drug therapies, blood parameters, and the clinical severity of each disease. The lack of this data can limit the power of the analysis. Finally, vaccination status for each included patient was not available, not allowing us to evaluate the effectiveness of vaccination against pneumococcal disease.

Therefore, our analysis can make an important contribution to the study of the characteristics of this disease in our region.

## 5. Conclusions

*Streptococcus pneumoniae* is a pathogen capable of contributing to the hospital burden in terms of both hospitalizations and intra-hospital mortality despite the existence of effective primary prevention tools such as vaccines. HDRs represent a valid and useful epidemiological tool for evaluating the direct impact of pneumococcal disease on the population and indirectly for evaluating the effectiveness of vaccination strategies and directing them.

## Figures and Tables

**Figure 1 vaccines-11-01324-f001:**
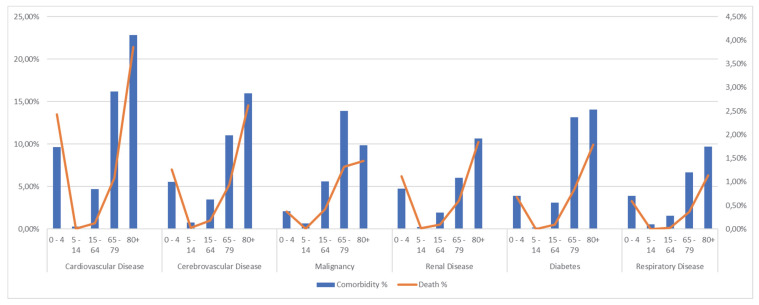
Overall distribution for five age classes (0 to 4, 5 to 14, 15 to 64, 65 to 79, and over 80) of comorbidity (left axis and blue bar) and in-hospital mortality (right axis and orange line). The graph represents the most common comorbidities in our sample, coded according to the Charlson index.

**Figure 2 vaccines-11-01324-f002:**
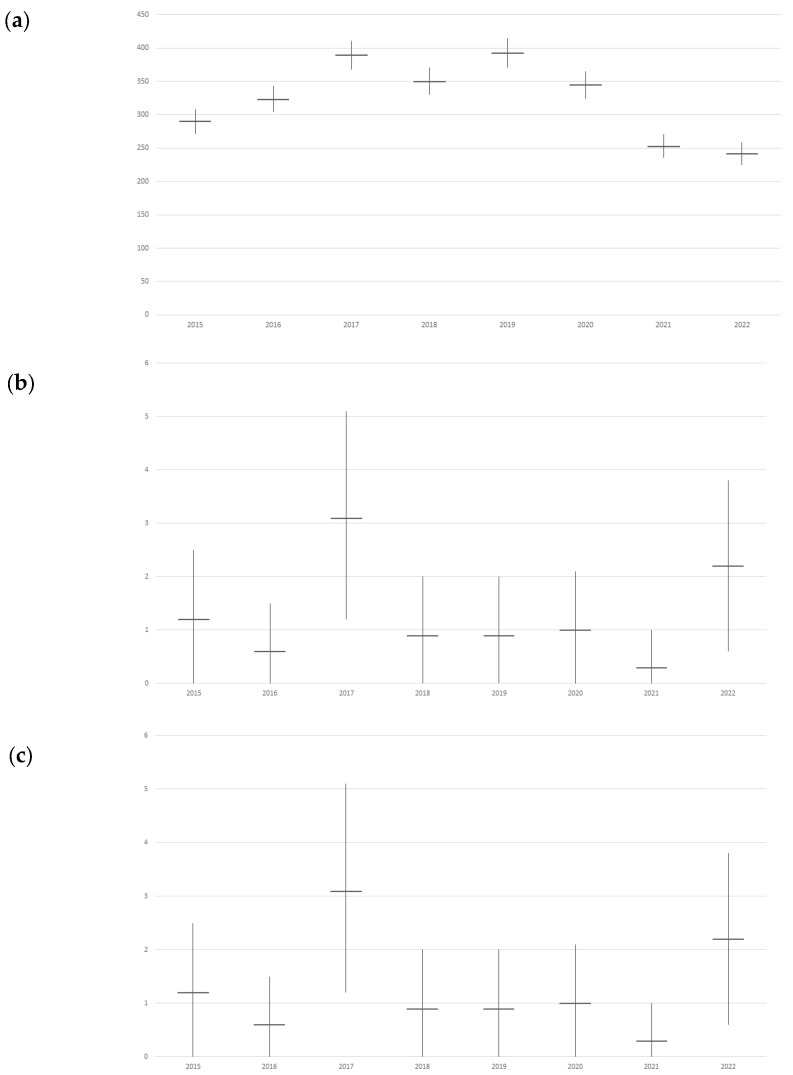
Admission rate trends per 100,000 Pescara province people for *Streptococcus pneumoniae*-related disease by year: (**a**) Estimated S.P. pneumonia admission rate trend; (**b**) S.P. meningitis admission rate trend, there was not the possibility of estimating unspecified cases as S.P.-related due the absence of these diagnosis; (**c**) Estimated S.P. bacteremia admission rate trend.

**Table 1 vaccines-11-01324-t001:** Hospital admissions from the Pescara Province from 2015 to 2022, overall and grouped by Pneumococcus-related diagnosis, SP Pneumoniae, Unspecified Pneumoniae (a), SP Meningitidis, SP Bacteremia, and unspecified Bacteremia (b). Each diagnosis set is further segmented by sex (male and female), age classes (0 to 4, 5 to 14, 15 to 64, 65 to 79, and 80 and above), year of discharge (2015 to 2022), number of comorbidities coded according to the Charlson index (no comorbidities, 1, and more than 1), and in-hospital mortality (yes or no).

(a)	Overall	%	All Pneumonia	%	S.P. Pneumonia	%	Unspecified Pneumonia	%
	288,110		14,506	5.035	7906	2.744	1673	0.581
M	140,699	49	8234	57	4351	55	1058	63
F	147,411	51	6272	43	3555	45	615	37
0–4	46,954	16	2508	17	1791	23	335	20
5–14	13,963	5	268	2	216	3	9	1
15–64	121,582	42	3651	25	1444	18	348	21
65–79	68,508	24	4157	29	2184	28	485	29
80+	37,103	13	3922	27	2271	29	496	30
2015	39,225	14	1205	8	842	11	268	16
2016	38,804	13	1230	8	990	13	143	9
2017	37,255	13	1468	10	1185	15	184	11
2018	36,961	13	1537	11	1158	15	230	14
2019	37,119	13	1570	11	1226	16	183	11
2020	30,166	10	2561	18	1056	13	172	10
2021	33,240	12	2619	18	779	10	144	9
2022	35,340	12	2316	16	670	8	349	21
0	192,088	67	7432	51	3539	45	898	54
1	59,573	21	4787	33	2809	36	549	33
2+	36,449	13	2287	16	1558	20	226	14
Y	13,434	5	3060	21	1686	21	457	27
N	274,676	95	11,446	79	6220	79	1216	73
**(b)**	**S.P.** **Mening.**	**%**	**S.P.** **Bacteremia**	**%**	**Unspecified** **Bacteremia**	**%**		
	33	0.011	88	0.031	5	0.002		
M	16	48	57	65	4	80		
F	17	52	31	35	1	20		
0–4	3	9	13	15	0	0		
5–14	1	3	5	6	1	20		
15–64	17	52	38	43	4	80		
65–79	10	30	23	26	0	0		
80+	2	6	9	10	0	0		
2015	4	12	5	6	0	0		
2016	2	6	6	7	0	0		
2017	10	30	6	7	0	0		
2018	3	9	1	1	1	20		
2019	3	9	24	27	2	40		
2020	3	9	13	15	0	0		
2021	1	3	16	18	1	20		
2022	7	21	17	19	1	20		
0	20	61	64	73	0	0		
1	12	36	17	19	2	40		
2+	1	3	7	8	3	60		
Y	3	9	8	9	1	20		
N	30	91	80	91	4	80		

**Table 2 vaccines-11-01324-t002:** Logistical analysis for in-hospital mortality. The independent variables included age, categorized into classes (with 14–65 as the reference group), sex (with male as the reference), invasive bacterial diseases, and comorbidities coded according to the Charlson index.

		Odds Ratio	IC 95%	*p*
**AGE**	0–4	4.786	4.494–5.098	<0.001
	5–14	0.153	0.101–0.234	<0.001
	65–79	2.868	2.698–3.049	<0.001
	80+	6.599	6.207–7.015	<0.001
	SEX	0.987	0.951–1.024	0.480
**Invasive Bacterial Disease**	SP Pneumonia	4.528	4.257–4.816	<0.001
	Unspecified Pneumonia	7.098	6.304–7.991	<0.001
	SP meningitidis	3.443	1.012–11.716	0.048
	Unspecified meningitidis	(omitted)		
	SP Bacteraemia	2.201	1.001–4.837	0.050
	Unspecified Bacteraemia	4.052	0.418–39.245	0.227
**Comorbidity** **(Charlson)**	Myocardial Infarction	1.140	1.024–1.269	0.017
	Chronic Heart Failure	2.333	2.221–2.451	<0.001
	Periferal Vascular Disease	1.217	1.083–1.367	0.001
	Cerebro-vascular Disease	2.483	2.360–2.611	<0.001
	Dementia	2.302	2.096–2.529	<0.001
	CPD	0.931	0.866–1.001	0.054
	Rheumatologic disease	0.567	0.433–0.741	<0.001
	Peptic Ulcer	1.520	1.187–1.947	0.001
	Mild Liver Disease	1.990	1.787–2.217	<0.001
	Diabetes without complication	1.104	1.039–1.172	0.001
	Diabetes complicated	1.113	0.921–1.344	0.267
	Plegia	1.234	0.920–1.654	0.160
	Renal Disease	2.051	1.933–2.176	<0.001
	Any malignancy	2.040	1.914–2.175	<0.001
	Liver Disease	3.666	3.394–3.959	<0.001
	HIV	(omitted)		

**Table 3 vaccines-11-01324-t003:** (**a**) Admission rate per 100,000 Pescara province people for all S.P.-related admissions and for S.P.-related diseases: pneumonia, meningitis, and bacteremia. (**b**) In-hospital mortality rate per 100,000 Pescara province people for all S.P.-related admissions and for S.P.-related diseases: pneumonia, meningitis, and bacteremia.

(a)	*Streptococcus pneumoniae* Admission Rate							
	All	IC 95%	Pneumonia	IC 95%	Meningitis	IC 95%	Bacteremia	IC 95%
2015	343.3	323.2–363.4	290.3	271.8–308.8	1.2	0–2.5	1.5	0.4–2.5
2016	388.4	366.9–409.8	323.6	304.0–343.1	0.6	0–1.5	1.9	0.4–3.4
2017	469.5	446–493.1	389.3	367.9–410.8	3.1	1.2–5.1	1.9	0.4–3.4
2018	443.9	421–466.8	350.5	330.2–370.8	0.9	0–2.0	0.3	0–0.8
2019	501.1	476.6–525.6	392.4	370.7–414.0	0.9	0–2.0	7.6	4.5–10.6
2020	446.1	423.1–469.1	344.8	324.6–365.1	1.0	0–2.1	4.1	1.9–6.3
2021	381.7	360.5–402.9	253.1	235.9–270.4	0.3	0–1.0	5.0	2.6–7.5
2022	361.4	340.7–382	241.9	225.0–258.7	2.2	0.6–3.8	5.3	2.8–7.9
**(b)**	***Streptococcus pneumoniae*** **In-Hospital Death Rate**							
	**All**	**IC 95%**	**Pneumonia**	**IC 95%**	**Meningitis**	**IC 95%**	**Bacteremia**	**IC 95%**
2015	61.3	52.8–69.9	45.9	38.5–53.2	0	-	0	-
2016	61.6	53.0–70.1	54.5	46.4–62.6	0	-	0	-
2017	77.2	67.6–86.8	73.1	63.7–82.4	0	-	0	-
2018	71.1	61.9–80.4	67.8	58.8–76.9	0.6	0–1.4	0.3	0–0.8
2019	76.6	65.7–84.7	76.4	66.9–86.0	0	-	0.6	0–1.4
2020	150.9	137.6–164.2	79.2	69.5–88.9	0	-	0.3	0–0.9
2021	127.5	115.4–139.7	63.7	55.1–72.2	0	-	0.6	0–1.5
2022	111.9	100.4–123.3	57.9	49.7–66.2	0	-	0.6	0–1.5

## Data Availability

Data were not available due to policy restriction of Abruzzo Region.

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
