# Peer review of "The Burden of Streptococcus pneumoniae-Related Admissions and In-Hospital Mortality: A Retrospective Observational Study between the Years 2015 and 2022 from a Southern Italian Province"

_vaccines, 2023, doi:10.3390/vaccines11081324_

Round 1
Reviewer 1 Report
The manuscript entitled “The burden of Streptococcus Pneumoniae related admissions and in-hospital mortality: a retrospective observational study between years 2015-2022 from an Southern Italian Provinc” is well written and nicely executed in term of experimentation. However, the quality of manuscript need to be strengthened by addressing the following concerns/queries:
1. Manuscript presentation needs to be further improved. Authors should discuss the strength as well limitations of the present study in the discussion section. The section “strength as well limitations” should be removed in order to avoid the ambiguity to the readers.
2. In figure 1, the similar death pattern is reported in cases of cardiovascular disease, cerebrovascular disease, renal disease, diabetes. Authors should add the justification for these observations.
3. The death rate is negligible for the age group 5-14. Authors should discuss these patterns of death rates.
4. Authors should have clarified the statement in page 7 “The most part of comorbidities caused a higher risk of in-hospital mortality, except for CPD (p=0.054) and complicated diabetes (p=0.267) and any -plegia (p=0.160).”
Author Response
- The manuscript entitled “The burden of Streptococcus Pneumoniae related admissions and in-hospital mortality: a retrospective observational study between years 2015-2022 from a Southern Italian Provinc” is well written and nicely executed in term of experimentation.
Reply: We thank the referee for the appreciation of our work.
- Manuscript presentation needs to be further improved. Authors should discuss the strength as well limitations of the present study in the discussion section. The section “strength as well limitations” should be removed in order to avoid the ambiguity to the readers.
Reply: Discussion section was improved as suggested by referee.
- In figure 1, the similar death pattern is reported in cases of cardiovascular disease, cerebrovascular disease, renal disease, diabetes. Authors should add the justification for these observations.
Reply: Discussion section was improved as suggested by referee.
- The death rate is negligible for the age group 5-14. Authors should discuss these patterns of death rates.
Reply: This point was discussed in Discussion section.
- Authors should have clarified the statement in page 7 “The most part of comorbidities caused a higher risk of in-hospital mortality, except for CPD (p=0.054) and complicated diabetes (p=0.267) and any -plegia (p=0.160).”
Reply: We modified the sentence. The meaning of this point was that the great part of comorbidities included in our evaluation were significantly associated to in-hospital mortality, with the exception of COPD (p=0.054), complicated diabetes (p=0.267) and any -plegia (p=0.160).
Reviewer 2 Report
The manuscript by Cedrone et al is Interesting paper that contributes confirmation of the patient groups most susceptible to pneumococcal infection. If access was available to patient records why not also comment on the pneumococcal vaccination status of those included in the study – the journal this paper is submitted to is Vaccines ?
Also, there are formatting and clarity issues with this paper that need addressing.
Introduction
Including line numbers would have helped the review.
Streptococcus Pneumoniae – please be consistent (and correct) in the manuscript as to how you cite and format species names - Streptococcus pneumoniae
Phrasing – use formal English, and proof read the manuscript.
Numerics – using 288.110 is confusing– you mean 288, 110 records were examined?
Methods
2.1. Coding of Streptococcus Pneumoniae—Hospital Admission
For the selection of admissions with or without directly specified etiology, the fol-lowing ICD-9-CM codes were used for the relative diagnoses. Codes 481, 482.9 and 041.2, 485 and 041.2, 486 and 041.2 were used.
To non-clinicians these codes would have little meaning, and be unclear to some clinicians. Explain the meaning of the codes used.
Results
Table 1 – numbers need to be on single lines, and the legend needs to explain the data more clearly.
Figure 1 needs to be better separated from Table 1 and this legend also needs to explain the data more clearly.
Table 2 has the same problem as Table 1- numbers need to be on single lines, and the legend needs to explain the data more clearly.
Figure 2– vertical axes lack legend as to what the data is showing.
M/s needs a thorough proof read - the English is understandable but needs polishing in places.
Author Response
Introduction
Including line numbers would have helped the review.
Streptococcus Pneumoniae – please be consistent (and correct) in the manuscript as to how you cite and format species names - Streptococcus pneumoniae
REPLY: Thanks for the precise considerations. The paper has been modified according to the indications of the reviewer
Phrasing – use formal English, and proof read the manuscript.
REPLY: The manuscript was revised according to the indications
Numerics – using 288.110 is confusing– you mean 288, 110 records were examined?
REPLY: the typo has been corrected
Methods
2.1. Coding of Streptococcus Pneumoniae—Hospital Admission
For the selection of admissions with or without directly specified etiology, the fol-lowing ICD-9-CM codes were used for the relative diagnoses. Codes 481, 482.9 and 041.2, 485 and 041.2, 486 and 041.2 were used.
To non-clinicians these codes would have little meaning, and be unclear to some clinicians. Explain the meaning of the codes used.
REPLY: The manuscript was corrected according to the indications and the pathologies were reported together with the ICD 9 coding
Results
Table 1 – numbers need to be on single lines, and the legend needs to explain the data more clearly.
Figure 1 needs to be better separated from Table 1 and this legend also needs to explain the data more clearly.
Table 2 has the same problem as Table 1- numbers need to be on single lines, and the legend needs to explain the data more clearly.
Figure 2– vertical axes lack legend as to what the data is showing.
REPLY: Thanks for the recommended fixes. The figures and tables have been corrected as indicated.
Round 2
Reviewer 2 Report
Thank you Authors for your replies. Some corrections have been made, but others not. Before I pointed out:
Streptococcus Pneumoniae – please be consistent (and correct) in the manuscript as to how you cite and format species names - Streptococcus pneumoniae
This has still not been done, and its continued lack affects credibility of the paper.
This time, Table 3 has numbers on different lines - the date (a) line.
The reason for not connecting pneumococcal disease to missing data on vaccine status I accept.
The English still needs a check on formatting and citation of species names.
Author Response
Streptococcus Pneumoniae – please be consistent (and correct) in the manuscript as to how you cite and format species names - Streptococcus pneumoniae
This has still not been done, and its continued lack affects credibility of the paper.
Reply: We thank the reviewer for consideration. Throughout the paper, the name has been formatted according to the indications.
This time, Table 3 has numbers on different lines - the date (a) line.
REPLY: We thank the reviewer for timely suggestion. The table has been formatted so that each number appears in a single line.
The reason for not connecting pneumococcal disease to missing data on vaccine status I accept.
REPLY: We thank the reviewer for accepting our work as valid